# ‘COVID Is Coming, and I’m Bloody Scared’: How Adults with Co-Morbidities’ Threat Perceptions of COVID-19 Shape Their Vaccination Decisions

**DOI:** 10.3390/ijerph20042953

**Published:** 2023-02-08

**Authors:** Leah Roberts, Michael J. Deml, Katie Attwell

**Affiliations:** 1School of Social Sciences, The University of Western Australia, Perth, WA 6009, Australia; 2Institute of Sociological Research, University of Geneva, 1211 Geneva, Switzerland; 3Division of Social and Behavioural Sciences, School of Public Health, University of Cape Town, Cape Town 7925, South Africa; 4Wesfarmers Centre of Vaccines and Infectious Diseases, Telethon Kids Institute, Perth, WA 6009, Australia

**Keywords:** comorbidities, vaccine hesitancy, COVID-19, Extended Parallel Process Model, mandates

## Abstract

Adults with comorbidities have faced a high risk from COVID-19 infection. However, Western Australia experienced relatively few infections and deaths from 2020 until early 2022 compared with other OECD countries, as hard border policies allowed for wide-scale vaccination before mass infections began. This research investigated the thoughts, feelings, risk perceptions, and practices of Western Australian adults with comorbidities aged 18–60 years in regard to COVID-19 disease and COVID-19 vaccines. We conducted 14 in-depth qualitative interviews between January and April 2022, just as the disease was starting to circulate. We coded results inductively and deductively, combining the Extended Parallel Process Model (EPPM) and vaccine belief models. Non-hesitant participants believed COVID-19 vaccines were safe and effective at mitigating COVID-19′s threat and subsequently got vaccinated. Vaccine hesitant participants were less convinced the disease was severe or that they were susceptible to it; they also did not consider the vaccines to be sufficiently safe. Yet, for some hesitant participants, the exogenous force of mandates prompted vaccination. This work is important to understand how people’s thoughts and feelings about their comorbidities and risks from COVID-19 influence vaccine uptake and how mandatory policies can affect uptake in this cohort.

## 1. Introduction

The COVID-19 pandemic has had far-reaching consequences, with some people facing greater risk from the disease than others. Adults with specific comorbid conditions, including obesity and immunocompromising conditions among others, are at increased risk of severe symptoms from COVID-19 infection [1]. Due to this risk, health officials in many countries prioritised these adults during the initial phases of COVID-19 vaccine programmes. In Australia, adults aged under 70 years with underlying medical conditions and disability were included in the second phase of the vaccine rollout, beginning in late March 2021 [2]. However, non-elderly adults with comorbidities are often not well-reached by other government-funded vaccination programs [3]. Hence, despite this group being a priority, there were challenges regarding uptake. These challenges were exacerbated after authorities declared that the Pfizer vaccine was preferred over the AstraZeneca vaccine in adults younger than 60 (8 April 2021) [4] and then 50 years of age (17 June 2021) [5] in a setting of low community transmission, due to risk of thrombosis with thrombocytopenia syndrome (TTS). In October 2021, authorities recommended that people aged 12 years and older who were severely immunocompromised receive an additional dose (third dose) [6]. In December 2021, priority groups that had received their vaccinations before June 2021 were eligible to receive a booster dose. At the time when the qualitative interviews were conducted for this study (January 2022–April 2022), four COVID-19 vaccines were available in Australia: Vaxzevria (henceforth AstraZeneca), Comirnaty (henceforth Pfizer), Spikevax (Moderna), and Nuvaxovid (Novavax). The latter was the last vaccine brand to become available (in January 2022) [7,8]. This meant that for much of the rollout, people aged under 50 were only recommended to receive or could only access mRNA vaccines, which was a source of concern for some regarding the new technology [9]. In all states across Australia, once eligible, people could receive vaccinations at no cost at a variety of locations including clinics led by General Practitioners (GPs) with funding and organisation from the federal government, as well as community pharmacies, and mass vaccination clinics run by the Western Australian Government.

### 1.1. Literature Review and Background

Given their susceptibility to severe illness from contracting COVID-19 and their prioritisation as a cohort for vaccination, it is important to understand how adults with co-morbidities relate to both the threat of COVID-19 and to COVID-19 vaccination. Increasing global attention and the ever-growing body of literature on vaccine hesitancy, vaccine acceptance, and vaccine refusal point to the complex reasons why people might not get vaccinated. Some of these factors are systemic, such as access barriers or poorly designed services that do not reach certain cohorts [10]. There are also attitudinal factors, including a lack of confidence in vaccines [11]. Vaccine hesitancy refers to people’s indecisiveness in decision making around vaccination [12], especially as it relates to concerns about safety, efficacy, or necessity of vaccination [13]. Researchers, having conducted a systematic review on global literature, argued that there is no ‘universal algorithm’ (p. 2155) and that the determinants of vaccine hesitancy are complex, context specific, and vary across time, place, and vaccine [14]. Not everybody who is vaccine hesitant refuses vaccines, and not everybody who refuses vaccines is ‘hesitant’—some are very clear in their decision to reject [10].

However, in the context of the fast-changing nature of the COVID-19 pandemic and COVID-19 vaccination programmes, it is essential to recognize that people’s vaccination attitudes can change over time. Carlson and colleagues qualitatively examined Western Australian adults’ intentions to receive COVID-19 vaccination following the announcement by the Australian Technical Advisory Group on Immunisation (ATAGI) that the Pfizer vaccine was preferred for adults in Australia aged <50 years due to the risk of TTS in younger people. Analysis of participants’ intentions to vaccinate identified four categories: (1) *acceptors*, who had no concerns about COVID-19 vaccine safety, efficacy, or access, (2) *cautious acceptors*, who had some concerns and preferred a particular vaccine brand but would accept what was offered, (3) *wait awhiles*, who sought more data, easier access, another brand, a greater perceived COVID-19 threat, or waited for mandates, and (4) *refusers*, who had no intention to vaccinate due to concerns about safety and/or efficacy [15]. Despite early studies in Australia indicating that as many as 14% of Australians were unsure about or unwilling to take COVID-19 vaccines prior to their development, the national two-dose coverage of 95.9% demonstrates that many subsequently changed their minds [16]. How vaccine decisions are made—and how they can change—is of key interest to the vaccination social science research field. Wiley et al., in developing and applying the concept of *vaccination trajectories* to refusers of childhood vaccines, argue that refusal is “not a static trait but rather the result of ever-changing experience and continual risk assessment” [17] (p. 1).

The COVID-19 pandemic has prompted people with co-morbidities to continuously assess risk in their everyday lives in light of changing epidemiological circumstances and scientific evidence. Existing global studies have shed light on this cohort’s thoughts, feelings, and experiences of the COVID-19 pandemic but have generally focused on adults over 65 years old [18,19,20]. Studies investigating younger adults with comorbidities have shown how this population felt more at risk at getting severely sick from COVID-19 [21]. Singh and colleagues investigated the health, psychosocial, and economic impacts of the pandemic on this cohort, finding that they had trouble accessing healthcare, experienced loss of employment, and experienced worsening of their health conditions [22]. This cohort was more likely to accept COVID-19 vaccines compared with their counterparts [23,24]. However, like the general population, younger people with comorbidities still had short- and long-term safety concerns, specifically regarding how their health conditions would interact with a COVID-19 vaccine; this was demonstrated in a study conducted in Victoria, Australia [25]. Concerns about vaccines interfering with or worsening comorbidities echo prior research into parents’ vaccine hesitancy and/or refusal of routine childhood vaccines. Some parents conceive of risk and vulnerability in their children based on their medical fragility and perceive that vaccines may exacerbate these problems or cause new ones. This “fragility framing” primes parents to assume that bad things will happen to their child’s health because they already have happened, and so the risk of adverse events following vaccination appears frighteningly likely [26].

In trying to make sense of how people think about disease and vaccine risk, scholars have employed the Extended Parallel Process Model (EPPM) developed by Witte (1992). The model was originally developed to provide a theoretical explanation for the responses people exhibit to fear-based health messaging, and it draws upon previous health behaviour models such as the Parallel Process Model and Health Belief Model [8]. The EPPM, which we elaborate further in the methods section, has been utilised in previous qualitative research to understand people’s risk perception and motivations for COVID-19 vaccination. For example, scholars in Poland found that both vaccinated and unvaccinated individuals perceived COVID-19 as a threat, but those who were unvaccinated were sceptical of the effectiveness of preventative health measures such as the vaccine [27]. Quantitative studies have also utilised the EPPM to explain COVID-19 vaccine uptake during the pandemic, finding similar results [28,29]. Prior to the pandemic, researchers used the model to understand vaccine uptake for conditions such as hepatitis B [30]. When considering vaccine uptake, the EPPM can be augmented by other tools that can measure how people think, feel, and act regarding vaccination. A range of tools have been developed and employed, particularly for understanding childhood vaccination decisions [31,32,33]. However, one straightforward approach for considering attitudes to vaccination is to ask whether people regard vaccines to be safe, effective, and necessary [34,35,36,37].

### 1.2. The Western Australian Pandemic and Vaccination Policy Context

The State of Western Australia (WA) experienced a unique context during the pandemic, with just over a thousand cases of COVID-19 in the first year [38]. Caseloads and community transmission remained extremely low until early 2022, when the Omicron variant spread through the community due to state borders reopening. A range of health measures were responsible for the earlier low caseloads, including the lengthy closures of international borders, mandatory two-week quarantine for overseas arrivals, and lockdowns when community cases were detected throughout 2020 and 2021. Such suppression methods differed across the country, particularly the closure of state and intra-state borders. By comparison to other states, WA experienced extended periods of state hard borders and strict intra-state border regulations [39].

Just over six months into the rollout, WA’s government began utilising vaccine mandates to drive uptake. From 8 August 2021, mandates were announced and subsequently implemented, initially covering key occupational groups (e.g., aged care, some healthcare, mining workforce jobs) [40,41]. By 31 January 2022, 75% of the workforce was required to have two doses of the vaccine in order to keep their jobs. [42]. Affected industries included the emergency services, supermarkets, schools, critical infrastructure, and restaurants. On 22 December 2021, the government added a requirement to receive a third dose within one month of being eligible [43]. At the beginning of the interview period for this present study (January 2022), WA borders were closed to interstate travellers, but from 5 February 2022, interstate travel was open. However, any travellers entering WA were required to have received two vaccine doses in addition to supplying several negative PCR tests [44].

Within this context, and in light of the prioritisation of people with comorbidities for COVID-19 vaccination due to the threat they faced from the disease, we sought to answer the following two research questions: (1) How do the thoughts and feelings about COVID-19 disease of WA adults with comorbidities affect their attitudes towards COVID-19 vaccines? (2) For these same individuals, how do their thoughts and feelings about COVID-19 vaccines inform and affect their uptake (or intended uptake)?

## 2. Materials and Methods

This research was conducted as part of the [blinded for review]. For this particular study of people with comorbidities, recruitment commenced in January 2022, although people could express their interest in participating from March 2021, when recruitment for the larger study commenced using media promotion, word-of-mouth, and snowballing. Those who were interested in participating signed up via an online REDCap survey [45,46], which collected demographic data and contact details, including if the individuals self-identified as having any health conditions (comorbidities) that make them more vulnerable to COVID-19. To capture the experiences of younger people, we sought to include only people under 60 years of age in this present study. We decided to interview this cohort once people with comorbidities were eligible for their booster dose of vaccination after December 2021, so we began contacting eligible individuals on 10 January 2022 following the festive break. The first author contacted prospective participants up to three times each by telephone and/or email to organise a face to face or telephone interview. Interviews were conducted between the 19 January 2022 and the 22 April 2022 and took approximately 60 min. Informed consent was obtained from all participants prior to the interview to participate in the research, including for publication of this paper. We purposively sampled for some individuals with comorbidities who indicated via the RedCAP survey that they had not received or were undecided about taking a COVID-19 vaccine and subsequent booster and/or had previously refused or delayed other vaccinations. We henceforth refer to those we purposively sampled in this way as “vaccine hesitant” participants. We ceased contacting and interviewing further potential participants once we had reached saturation of both hesitant and non-hesitant people with comorbidities.

We conducted semi-structured interviews to ask participants about their COVID-19 vaccination attitudes and experiences; a detailed question guide can be referenced in the study protocol [47]. Additional questions for people with comorbidities focused on their perceptions of COVID-19 and vaccination risks due to their conditions. Other questions investigated their perceptions of government policies, the medical community, and the attitudes and behaviours of those around them during the pandemic. Interviews were audio recorded and transcribed verbatim, then coded by the lead author using NVivo 20, in regular consultation and meetings with the other authors. Inductive coding was undertaken to draw out emergent themes [48]. Then, deductive methods informed theoretical coding on several fronts.

The first round of deductive coding divided participants into five categories based on a modified framework from Carlson et al. [15] to understand individuals’ COVID-19 vaccine intentions. This framework, as described in this article’s introduction, categorised participants as *acceptors*, *cautious acceptors*, *wait awhiles* or *refusers* [15]. Given that the framework was developed before the full rollout commenced, we adapted it to reflect vaccine status, not just intentions, and to include the category of *coerced acceptor*. These were people who had reluctantly received the vaccine only due to WA’s employment-related vaccine mandates, as described in the Introduction. For the other categories, we classified *acceptors* and *cautious acceptors* as those who had sought to receive their first two vaccine doses in the recommended timeframe for their vaccination group, as well as those having received or who had made an appointment for a booster, in line with recommendations at the time. The difference between the two groups lay in the latter’s minor concerns about being vaccinated. *Wait awhiles* had deliberately delayed their first vaccine doses. *Refusers* had not been vaccinated against COVID-19 or had commenced but deliberately ceased their recommended course of vaccines (See Figure 1).

Once we had classified all participants into one of these five categories, the lead author conducted further deductive analysis based on two key constructs used in health and vaccination research and outlined in our introduction to this article. First, we utilised aspects of the EPPM as a heuristic [49]. Following other scholars of vaccination mentioned in our literature review [28,29], we utilised the model to consider the ways that participants might respond practically not just to fear *messages* (as the model was originally designed) but to an objective threat of contracting and having severe health consequences from COVID-19. According to the EPPM, an individual perceives *threat* based on its *severity* (i.e., perceived seriousness of threat) and the individual’s *susceptibility* (i.e., perceived probability of experiencing the threat). Then, *efficacy* refers to the effectiveness of following recommended responses to addressing the threat. Efficacy is broken down into two components: *self-efficacy* (a person’s ability to perform the recommended response to the threat) and *response efficacy* (their beliefs about the whether the response will effectively prevent the threat). The model reports them in this order, but we have reversed them in the presentation of our results to better suit the chronology of vaccine decision making.

To deepen our analysis regarding the perceived efficacy of taking the vaccine, we considered people’s attitudes towards vaccination’s *safety*, *efficacy*, and *necessity* [34,35,36,37]. As ‘necessity’ was covered by people’s attitudes towards COVID-19 disease severity and their susceptibility, we focused here on attitudes towards vaccine *safety* and whether people regarded the available vaccines to be sufficiently *effective*. We considered these aspects of a vaccination journey in an abstract sense (Figure 2) and populated the journeys for each participant to demonstrate how COVID-19 disease fears inform (or do not inform) vaccine uptake, as mediated by attitudes about vaccine safety and efficacy. At the end of our results section, we report the journeys of three participants who best represented archetypal journeys of our sample.

## 3. Results

We interviewed 14 participants, 4 of whom were vaccine hesitant. Participant ages ranged from 21 to 60 years old (average ~39 years), nine were female, four were male, and one preferred not to disclose gender details. Participants reported a range of comorbidities including Type 1 and 2 diabetes, high blood pressure, autoimmune conditions, and asthma. Table 1 includes our categorisation of individuals according to the modified vaccination status model (Figure 1) and notes their vaccine dosage based on the two initial doses in the vaccination schedule and the one scheduled booster at the time of study. Seven participants were *Acceptors* (no concerns about vaccine safety or efficacy), two were *Cautious Acceptors* (minor concerns), one was a *Wait Awhile* (who waited for more information about how the vaccine would affect her, and to feel more confident), two were *Coerced Acceptors* (did not want the vaccine but took it due to policy levers), and two were *Refusers* (unwilling to have the vaccine due to perceived lack of vaccine safety, necessity, and efficacy).

### 3.1. Perceiving the Threat: Severity Perceptions of COVID-19

Considering the participants’ interview responses in light of the EPPM, we saw them first appraising the threat of the hazard, which in this case meant contracting and becoming severely unwell from COVID-19, including potentially acquiring long COVID-19.

There were varying degrees of concerns amongst participants about contracting COVID-19. Dawn (59, Asthma, Acceptor, Triple Dosed) explained the risk she felt of contracting COVID before she was fully vaccinated,


*“I’m fully vaccinated and boosted at this stage, but I also suffer from asthma and have done for a long time. So, I knew that if I didn’t get vaccinated and I caught it, it would have killed me.”*


Participants used their comorbidities as a point of comparison to people around them without health conditions. They also drew on their previous experiences of acute ill health to consider their specific risks regarding contracting COVID-19. For example, Jess (22, Autoimmune condition, Acceptor, Triple Dosed) shared,


*“I would say that I’ve been pretty cautious about the virus… I know what it’s like to get sick and not fully recover, and then I also I don’t think my body is a as equipped to fight the virus as perhaps another person my age.”*


Others were concerned about the potential severity of the virus adding to their pre-existing conditions, as several participants noted. Liz (46, Type 1 Diabetes, Acceptor, Triple Dosed) shared, “*It’s not the virus itself so much as possible complications*”. Redgum (21, MADD, Acceptor, Triple Dosed) explained, “*I don’t want it because I’ve just got far too may medical conditions already*”. Phoenix (45, Acceptor, Triple Dosed) said, “*My husband and I both have illnesses that we’re not sure how they would co-react with COVID*”.

Some participants shared that long COVID post-infection was a key concern. Anna (High blood pressure, 24, Acceptor, Triple Dosed) shared, “*Long COVID. That’s the part I’m afraid of, and that concerns me greatly, and I will do anything I can to avoid getting COVID, specifically because of long COVID*”. Gillian (23, Type 1 diabetes, Cautious Acceptor, Triple Dosed) concurred, “*The whole idea of it being permanent or being six months instead of, you know, a one-week or two-[week] illness”* rendered long COVID “*another disability*” which would “*exacerbate the conditions I already have, not necessarily viewing it as something completely separate”* and *“make life harder*”.

Vaccine hesitant participants also held some concerns about contracting COVID-19. However, they often presented this retrospectively. For example, Charlie (30, Neurological Conditions, Coerced Acceptor, One Dose) said that in the early days of the pandemic, he was quite concerned, whereas Sophie (43, Lupus, Refuser, No Vaccinations) shared that she was less worried at the time of the interview than she had been previously.

However, vaccine hesitant participants often depicted concerns about the virus as ‘*over-inflated*’, indicating lower levels of fear than the non-hesitant participants. Stewart (60, Obesity, Coerced Acceptor, One Dose) shared that despite belonging to the age and lifestyle category that would mean he is at greater risk of dying from COVID, he was not concerned by the virus*, “I still can’t be afraid of it. It’s like being afraid of driving down the road”.* He further elaborated his reasoning by arguing that deaths from COVID-19 were relatively low when compared with other causes of death: *“In a global or country effect of how many people are dying, it’s almost inconsequential”.* Sophie (43, Lupus, Refuser, No Vaccinations) noted that the Omicron strain is less severe than earlier variants and that across the world, deaths and hospitalisations are decreasing, which she did not attribute to vaccination. *“The risk factors [of getting severely sick from COVID-19] are becoming more apparent and I don’t think I’ve got many of those. I’m not so worried about it”.*

The same vaccine hesitant participants were similarly unconcerned about long COVID. Many thought there was not enough information about the condition. As Molly (49, Autoimmune Disorder, Refuser, No Doses) explained,


*“The research on long COVID isn’t clear, is it? I don’t think they’re in a position yet because there aren’t any longitudinal studies to be clear about long COVID. So I have no concerns for long COVID more than I do for any other infectious disease.”*


To summarise, non-hesitant participants held concerns about contracting COVID-19 and the severity of the infection they could have, particularly with their pre-existing conditions. By contrast, although hesitant participants did share some earlier worries, they also believed the risk was far less severe and were not more concerned about COVID-19 than they were about other viruses.

### 3.2. Perceiving the Threat: Susceptibility Perceptions of COVID-19

The second element of perceived threat is susceptibility, which Witte (1992) defines as an “individual’s belief about his or her chances of experiencing the threat”. Thus, we wanted to understand how all participants perceived their chances of contracting COVID-19. For non-hesitant participants, even though they were worried about catching COVID-19, they did not feel that they were particularly suspectable to the virus when we interviewed them because WA’s hard border policy was keeping them relatively isolated from the virus. Joan (34, Acceptor, Triple Dosed) felt safe and was optimistic that she was not going to be sick, *“We were isolated from everybody else. We were not going to be probably as affected…We’re doing the right things as well in terms of how we’re managing it by…keeping things closed”.*

At the time of the interviews, WA’s hard border was being lifted after two years. Participants saw their chance of contracting COVID-19 would be significantly higher once the border was open. As Boris (33, Asthma, Cautious Acceptor, Triple Dosed) explained, “*It is coming around quite quickly… I’m bloody scared, you know”.* Gillian (23, Type 1 diabetes, Cautious Acceptor, Triple Dosed) further shared how in 2022 things changed for her, “*I think I’ve kind of been a little bit more frightened, ‘cause I know it’s more of a possibility [that I’ll contract COVID-19]”.*

However, even though they did not feel particularly at risk at the time, participants felt susceptible to being infected with COVID-19 because they believed it had higher rates of transmissibility compared with other infectious diseases, such as influenza, and that people could pass it on whilst appearing to be asymptomatic. Joan (34, Acceptor, Triple Dosed) said,


*“I think that I’ve got a higher chance of getting COVID than I do flu and pneumonia because I don’t really hang around a lot of people who are sick with the flu. But, COVID is another thing altogether.”*


In contrast to the other participants, vaccine hesitant participants minimized their susceptibility to catching COVID-19 by emphasising the protection offered by living in WA. Stewart (60, Obesity, Coerced Acceptor, One Dose) explained, *“I live in WA… I spend 95% of my life at home, my risk of being exposed to that virus in WA, where up until recently, we basically didn’t have it at all”.* These participants did not, however, express concerns about the border opening, and Sophie (43, Lupus, Refuser, No Vaccinations) expressly declared that it should open.

Thus, participants had differing levels of concerns about the contracting COVID-19 due to WA’s hard border policy, as prior to 2022, there had been very few cases in the state. However, with the border being lifted and the belief that COVID-19 is more transmissible than other diseases, participants felt more susceptible to contracting COVID. Vaccine hesitant participants did not share the same level of concern.

### 3.3. Responding to the Threat with Vaccination: Response Efficacy

According to the EPPM, after the assessment of the perceived threat, individuals then evaluate the efficacy of the recommend response, which here refers to taking COVID-19 vaccines. In the EPPM, response efficacy refers to an individual’s belief as to whether a response effectively prevents the threat. In this context, this concerns participants’ evaluation of COVID-19 vaccines as being an appropriate way to prevent COVID-19 infection. Vaccines would be appropriate if people saw them as safe, effective, and necessary. If participants saw the disease as a threat, as we have described above, then we took the necessity as a ‘given’. However, the vaccines would also need to be regarded as safe and able to effectively prevent serious illness.

When participants were asked about their thoughts on receiving a COVID-19 vaccination, the majority reported feeling very positive about receiving or having had the vaccination. For some, the decision was a ‘no-brainer,’ and their comorbidities did not make them feel as though they may respond badly to vaccination. The majority of our participants stated that they did not feel more at risk from the vaccines than the general population. They expressed trust in medical professionals, particularly as they interacted with them to manage their conditions. For example, Phoenix (45, Acceptor, Triple Dosed) explained, “*We trusted what the doctors told us and what the GPs told us, and that was that it’s safe. I wasn’t scared to take the vaccine. That’s for sure. I just trusted what we were told”.*

Some participants did have concerns about the vaccine, including the effect it would have on them and their conditions, how quickly it was produced, and the effectiveness, but all felt reassured by medical professionals. For example, Liz (46, Type 1 Diabetes, Cautious Acceptor, Triple Dosed) said,


*“I did actually speak to my endocrinologist and say, ‘Look, you know, again that thing of knowing what immunity … autoimmune stuff is triggered by viruses. What is the chance that this is gonna trigger something?’ And she said, ‘Oh, it’s possible, but it’s that risk versus, you know, which risk is higher kind of thing’ … But, you know, I’m very vaccine positive so, you know, I’m willing to trust the experts on that.”*


Similarly, Gillian (23, Type 1 diabetes, Wait Awhile, Triple Dosed) shared she *“kind of”* thought she could have an adverse reaction but received reassurance from hearing that other people with type 1 diabetes were fine after receiving the vaccine.

By contrast, vaccine hesitant participants said that the vaccine posed more of a risk to them than COVID-19 disease, despite some also having concerns about contracting the disease. For example, Sophie (43, Lupus, Refuser, No Vaccinations) explained, “*I am worried about catching [COVID] as well, but I’m more worried about the vaccine, with my health issues”.*

Others also expressed concerns about vaccine efficacy as well as vaccine safety. Molly (49, Autoimmune Disorder, Refuser, No Doses) said, *“The risks of having an experimental drug were greater to me than the risks of contracting Coronavirus”.* She added, *“I don’t see that current strategies for managing COVID actually do provide the level of protection that they claim”.* Charlie (30, Neurological Conditions, Coerced Acceptor, One Dose) linked his reticence to the vaccine’s ‘*element of the unknown*’, although he did not ‘*personally believe that there’s likely to be serious long term adverse consequences’.*

In summary, the majority of participants felt the vaccine was safe and effective in mitigating the threat of COVID-19; however, vaccine hesitant participants perceived the vaccine as more of a threat than contracting COVID-19.

### 3.4. Responding to the Threat with Vaccination: Self-Efficacy

The second component of efficacy is *perceived self-efficacy*, which refers to an individual’s ability to perform the recommend response. Non-hesitant participants all saw vaccination as something that they could do and wanted to do, and in fact they described the vaccine as a way of exercising agency and control to prevent the risk of COVID-19 disease.

For example, Polly (55, Type 2 Diabetes, Acceptor, Triple Dosed) expressed, *“It was a great relief when I could say that I was fully vaccinated”.* Jess (22, Autoimmune condition, Acceptor, Triple Dosed) shared, “*I was actually pretty excited. It felt like I was able to do something proactive to protect myself and increase my changes of coming through this pandemic alright. It made me feel a lot safer”.*

However, Gillian (23, Type 1 Diabetes, Wait Awhile, Triple Dosed) shared that doubts about vaccine safety posed a barrier for her and said that she ‘*procrastinated’* about getting the vaccine: *“I think part of that was because I was like, ‘I wonder what’s gonna happen? Like is everyone gonna be okay? They haven’t done fifteen years of safety trials. Surely, that’s gonna be an issue’”.*

The two *Coerced Acceptors* only chose to get vaccinated because of vaccine mandates. Charlie (30, Neurological Conditions, Coerced Acceptor, One Dose) was very angry about the mandates, describing them as *‘complete overreach’* and *‘very, very, very offensive*’. He identified as someone engaging in *‘civil disobedience in the face of authoritarianism’.* Vaccine mandates made Charlie feel much more negatively about being vaccinated, but they also made him take the vaccine; at the time of the interview, Charlie had been placed been on leave without pay for two months and was in the process of getting fully vaccinated so that he could return to his job in WA’s mining sector. Stewart (60, Obesity, Coerced Acceptor, One Dose), who had begun a course of Novavax vaccinations, explained the influence of the employment mandates and the consequences of vaccination-related travel restrictions: “*I’d already decided a long, long time ago that I would get a Novavax... purely on the basis ‘cause I want to go back to work… I thought the only way I can go and visit my children is if I get vaccinated”.* Stewart believed that his adult children, who lived on the other side of the country, had experienced adverse reactions to the vaccines, and so he feared them facing potential booster requirements for travel to visit him. Stewart preferred to get himself vaccinated instead so that he could be the one visiting them. He was more comfortable with Novavax, as he believed it was similar to the influenza vaccination which he had previously had.

The two *Refusers* who chose not to vaccinate and did not share any intention of vaccinating in the future. Thus, the majority of participants in our sample did get vaccinated (or were in the process of doing so) and received booster shots, including two who felt they were coerced into making that decision.

### 3.5. Vaccination Journeys: How Responses to Fear and Vaccine Beliefs Shape Decisions

By tracing each participant through the phases we have described here and laid out in Figure 2, we have identified three illustrative vaccination or non-vaccination journeys that can stand in as ideal-types. Although we expected to find differences of perspective amongst participants due to place of birth, age, and gender, overall non-hesitant participants shared similar vaccination journeys. Here, we recount the journeys of Polly, Charlie, and Molly.


*Polly—a quintessential vaccine acceptance story*


Polly was afraid of catching COVID because of her comorbidities. She also believed that she had an increased chance of experiencing a severe outcome from COVID-19 and not recovering as well as other people might. These *fears of the disease* predisposed her to see the COVID-19 vaccines as *necessary*. Polly further believed that the vaccines had sufficient levels of *safety* and *effectiveness* to orient her towards accepting them. Polly therefore demonstrated *response efficacy* in believing that the vaccines could address the threat that COVID posed. She also demonstrated self-efficacy in seeing vaccination as a step she could take in addressing that threat. Polly’s journey is illustrative of a straightforward vaccination journey.


*Molly—a quintessential vaccine refusal story*


Molly’s non-vaccination journey was informed by a lower fear of the disease than that shared by Polly and other non-vaccine hesitant participants. Molly’s early fears of the disease had waned during the course of the pandemic, and instead she had come to see the vaccine as the more frightening prospect. Thus, with muted fears of the disease, Molly did not see the vaccine as *necessary*, and more importantly, she did not believe it was sufficiently *safe*. For Molly, the vaccines failed at the level of *response efficacy* and *self-efficacy.* She did not see that they could actually solve the threat of COVID-19, which she believed was not that significant anyway, due to her safety concerns. As a result, Molly refused the vaccines.


*Charlie—a story of external coercion shaping vaccine decisions*


Charlie’s journey generally started out resembling that of Molly. He did not see COVID-19 disease as particularly threatening and therefore did not see vaccines as *necessary*. He also did not believe they were *effective*, and although he did not have significant safety concerns, he believed that mandates were inappropriate in the context of novel vaccines. Where Charlie’s journey deviates from that of Molly is that he was compelled to vaccinate because of the WA Government’s mandatory vaccination policy. This policy did not change the way that Charlie thought or felt about any of the factors that drove his preference not to vaccinate: he was still not afraid of COVID-19 and remained somewhat concerned about the vaccines’ safety. However, importantly, Charlie was highly resentful of the policy that required him to choose between his job and his beliefs. However, the mandates exerted a powerful force on Charlie, reorienting his journey and resulting in him becoming vaccinated. We illustrate this in Figure 3.

## 4. Discussion

In this study of people with comorbidities in Western Australia, we have traced the stages of their vaccination journeys using a model informed by the EPPM and the constructs of vaccine safety, efficacy, and necessity.

Our analysis of aggregated themes—as well as individual participants’ journeys—demonstrated that fear or lack of fear of the disease was an important shaper of people’s vaccination outcomes, as discussed in previous literature [23,24]. Participants evaluated the threat of contracting and becoming severely ill from COVID-19 in terms of severity and susceptibility, as the EPPM suggests. Similar to the findings of Wolf et al., our study found that those who were concerned about becoming severely ill or developing long COVID believed they were susceptible to contracting COVID and evaluated it as a threat [21]. People’s comorbid conditions appeared to play an important role, with many believing that their pre-existing health conditions would increase the severity of the disease and its long-term impacts. Participants’ experiences of living with poor health or health concerns seemed to prime them to think about COVID-19 risk in these ways. Despite most participants perceiving a low risk of contracting COVID due to WA’s border closures, many shared that the border reopening would change this. Furthermore, as many other countries had begun their vaccine rollout prior to WA, this could have influenced participants to feel excited and reassured that the vaccines were safe.

Although vaccine hesitant participants had concerns about COVID-19, these concerns were less severe, and some reported them diminishing over time. Vaccine hesitant participants did evaluate COVID-19 as a threat in varying degrees, particularly at the beginning of the pandemic. These concerns may have diminished as participants came to see vaccines as the greater threat.

A similar divergence occurred between hesitant and non-hesitant participants regarding the perceived efficacy of the COVID-19 vaccinations for mitigating the threat of the disease. The participants who believed that the threat of disease was severe thought the vaccines were necessary, believed they were effective at preventing serious illness, and had confidence in their safety. They consequently took the vaccines. Some vaccine accepting participants, whom we classified as *Cautious Acceptors* or *Wait Awhiles*, did worry about the vaccines’ safety, echoing the findings of Kaufman et al. [25]. This meant that they had to weigh up the relative risk of the disease versus the vaccines. Overwhelmingly, these participants were reassured by their medical professionals that the vaccines were a safer bet.

By contrast, vaccine hesitant participants took different and varying journeys. This is reminiscent of the work of Wiley et al., which found that vaccine hesitant and refusing parents had various vaccination trajectories for their children’s vaccinations, with study participants’ vaccination journeys involving continual risk assessment over time [17]. Contrary to their research, which focused on childhood vaccination, our study provides novel insight into vaccination risk assessment among adults 18–60 years of age with comorbidities in the context of the COVID-19 pandemic. This, until now, is a research area that has received surprisingly little attention. When evaluating the efficacy of the vaccine in mitigating the threat of the disease, vaccine hesitant study participants felt that the risk of the vaccine was greater than the threat of contracting COVID. These participants reported becoming less concerned about COVID-19 over time. This corresponds to the timeline of the virus being initially terrifying (reports from Wuhan and Italy, for example), then Western Australians feeling protected from it (via border closures and lockdowns), and finally governments recommending and mandating vaccines. By this time, people may have felt shielded from the virus or less concerned about it and some focused their concerns on the vaccines instead. It is possible that vaccine hesitant participants engaged in motivated reasoning here. Motivated reasoning occurs when people search for evidence to support their gut feelings or attitudes, evidence that “reinforces the conclusion that one is motivated to reach” [50] (p. 308). By the time that vaccine hesitant participants had become convinced that the vaccines were unsafe or ineffective—and especially once they were angry at the perceived injustices of the government mandating them—they may have adjusted their perspectives on the threat of the disease in order to better align with the fact that they did not want to vaccinate.

It was striking that mandates operated as an external force on half of the vaccine hesitant participants and that these individuals chose to get vaccinated despite not altering their perceptions about COVID-19 or the vaccines. Rather, the costs of not vaccinating were too high for them to bear. The anger expressed by Charlie, in particular, indicates that mandates may pose costs later, generating psychological “reactance” and orienting people away from voluntary vaccination programs [51]. This may be seen at an aggregate level: WA still has the highest third-dose coverage of any Australian state [16], reflecting the period that the mandate required workers to take this third dose. However, the mandate was removed on 10 June 2022 for most industries [52] and the fourth dose was never mandated. WA has some of the lowest fourth dose coverage in the country [16]. Of course, there are many factors behind this, including different experiences of disease outbreaks and infection across the country during 2022, and people’s evolving perceptions of how to attain immunity from new strains in a context of continual community transmission. However, it is clear that participants such as Stewart and Charlie would not be having any more doses than required under WA’s mandates. The journeys of Stewart and Charlie do not fit with the traditional process of considering threat and response, as expressed in the EPPM. For these participants, despite their evaluation of the vaccine as more of a threat than contracting COVID-19, the external coercion of the mandates influenced them to take the vaccine. In relevant settings, future studies of vaccine decision making should aim to incorporate and further investigate the external motivation that mandates can produce.

The purposive sampling of non-elderly adults (aged 18–60 years old) with comorbidities provides important insight into a priority group for COVID-19 vaccination services. Contrary to what may have commonly been assumed in a purely rational risk-assessment biomedical approach, this ‘invisible’ risk-group did not categorically accept COVID-19 vaccination, despite increased objective risk to severe COVID-19 health consequences. There were individuals in our sample who did not immediately vaccinate in order to mitigate their risk of COVID-19 illness. Some ‘felt the fear but did it anyway’, some waited awhile, some were coerced into vaccination by mandates, and some refused. Vaccine hesitant individuals had other risks in mind and expressed concerns about vaccine safety, efficacy, and necessity. Some *Cautious Acceptors* and the *Wait Awhile* participant thought about their vaccination decisions by couching them within a ‘fragility framing’ [26]. In other words, some were concerned that their comorbid conditions may put them at higher risk of experiencing severe COVID-19 vaccination adverse events. Matters were then complicated by the policy lever of vaccine mandates, which, instead of alleviating concerns, increased perceptions of vaccine risk or angered certain participants. This last point is of particular import for vaccination policy considerations. Not all participants considered only health-related threats in their risk assessment. Moreover, this risk assessment was not statistically based or ‘rational’ from a biomedical perspective but instead drew upon participants’ considerations of the *non-medical* risks of non-vaccination. For example, vaccine mandates as a policy lever introduced *social* threats for vaccine hesitant participants, as these threats touched upon their ability to travel to see loved ones and on participants’ livelihoods.

This study has some limitations. Its qualitative design precludes us generalizing findings to the wider population of WA and to other contexts with different epidemiological and political realities. Our recruitment methods for interviews resulted in participants with high English proficiency and living in metropolitan Perth. However, within our sample, we aimed to interview a variety of ages, genders, and people with different conditions and to speak to people who were hesitant or refusing as well as those who had been fully vaccinated. Furthermore, we specifically focused on an ‘invisible’ group of people with comorbidities who were not also elderly; this group is often under-served by both comorbid vaccination programs and by researchers. Additional limitations include that people who agreed to participate in the research may have held stronger vaccination views (either in favour or opposed) compared with those who did not sign up. Furthermore, we had few participants who were outright vaccine refusers, due to Western Australia’s high vaccination rate of over 95% limiting the pool from which we could recruit. Relying on self-identification of comorbidities and not insisting that people disclose them to us, we did not medically verify that all participants fit into the comorbidity COVID-19 vaccination group in Australia.

Vaccination social science research will need to stay abreast of developments such as uptake of future dosages of existing or new COVID-19 vaccines, and it will be important to focus, as we have, on non-elderly adults with comorbidities because they will once again be priority groups for protection. Future research could more formally mobilize the EPPM in its entirety to understand people’s responses to vaccination threats and risks, going beyond our use of it as a coding framework and heuristic.

## 5. Conclusions

This research provides an important perspective from adults with comorbidities and their thoughts and feelings about COVID-19 disease and vaccination. The Extended Parallel Process Model was a useful heuristic for analysis in the context of West Australia in which, unlike many other settings, COVID-19 border policies allowed for mass vaccination in anticipation of widespread coronavirus infection. As an analytic frame, we combined the EPPM with vaccine belief models, which allowed us to produce and analyze a repertoire of factors influencing how adults with comorbidities evaluated risks around COVID-19 disease and made subsequent vaccination decisions. This included valuable concepts from the EPPM (*threat, severity, susceptibility, response efficacy, self efficacy*) and vaccine belief models (*necessity, safety, efficacy*). Adults with comorbidities in Western Australia have a unique perspective in how they were thinking about the risk of COVID-19 disease and vaccinations because there was a very low threat of contracting COVID-19 at the beginning of the pandemic. By tracing these adults’ vaccination or non-vaccination journeys, we saw how their consideration of health threats were highly contextual and therefore changed over time. Furthermore, their vaccination decisions were influenced by not only disease threat; policy factors, such as border closures, vaccination requirements for travel, and mandates linked to employment were important deciding factors for some.

## Figures and Tables

**Figure 1 ijerph-20-02953-f001:**
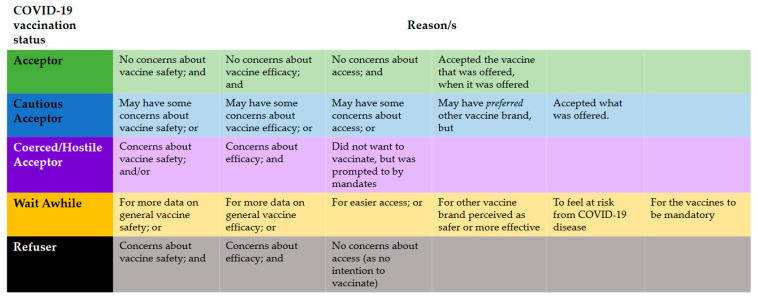
The recommended course of vaccines.

**Figure 2 ijerph-20-02953-f002:**
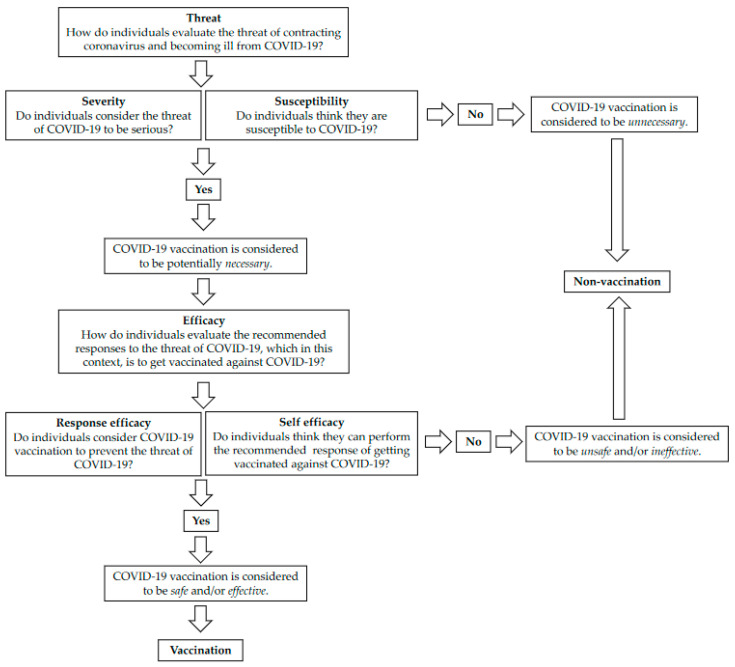
Vaccination journey in an abstract sense.

**Figure 3 ijerph-20-02953-f003:**
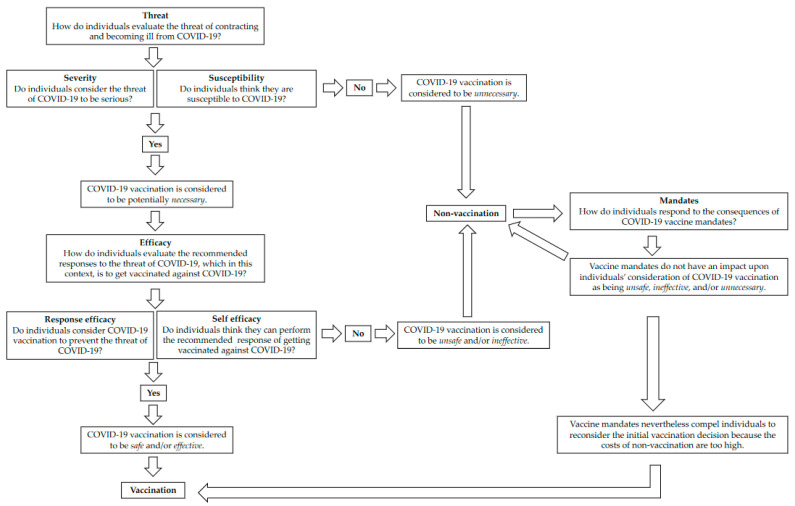
Patient journey and resulting in becoming vaccinated.

**Table 1 ijerph-20-02953-t001:** Demographic details of participants.

Vaccination Status and Intention	Pseudonym	Comorbidity (Self-Identified)	Gender	Age
Acceptor, Triple dosed	Anna	High blood pressure, obesity	F	24
Acceptor, Triple dosed	Redgum	MADD (multiple acyl-CoA dehydrogenase deficiency—rare disease)	M	21
Acceptor, Triple dosed	Joan	Did not disclose specific comorbidity	Prefer not to say	34
Acceptor, Triple dosed	Jess	Autoimmune condition	F	22
Acceptor, Triple dosed	Dawn	Asthma	F	59
Acceptor, Triple dosed	Phoenix	Did not disclose specific comorbidity	F	45
Acceptor, Triple dosed	Polly	Type 2 Diabetes	F	55
Cautious acceptor, Triple dosed	Liz	Type 1 Diabetes	F	46
Cautious acceptor, Triple dosed	Boris	Asthma	M	33
Wait Awhile, Triple dosed	Gillian	Type 1 Diabetes	F	23
Coerced acceptor, One dose	Stewart	Obesity	M	60
Coerced acceptor, One dose, second dose booked	Charlie	Chronic neurological conditions	M	30
Refuser, No doses	Molly	Autoimmune disorder	F	49
Refuser, No doses	Sophie	Lupus and other conditions	F	43

## Data Availability

Not applicable.

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
