# Peer review of "‘COVID Is Coming, and I’m Bloody Scared’: How Adults with Co-Morbidities’ Threat Perceptions of COVID-19 Shape Their Vaccination Decisions"

_ijerph, 2023, doi:10.3390/ijerph20042953_

Round 1
Reviewer 1 Report
Dear authors,
It was a pleasure reading your article, and I have no comments to add.
All the best!
Additional Comments:
This research investigated the thoughts, feelings, risk perceptions, and practices of Western Australian adults with comorbidities aged 18-60 years in regard to COVID-19 disease and COVID-19 vaccines.
It is extremely well-written research, and I have no comments to make on the project. It is certainly worth publishing as the topic is original and relevant to the field and addresses the gaps in current understanding.
The study utilised qualitative analysis and interestingly reports that people who were non-hesitant towards the vaccine also believed that COVID-19 vaccines were safe and effective. Therefore, the findings are of public health interest, and mandates that the practitioners and health providers spread the awareness on the safety related to the vaccines so that more people are convinced in taking the vaccination doses.
Author Response
Thank you for your feedback.
Reviewer 2 Report
I appreciate having the opportunity to review the article. There is much merit in a COVID perceptions study that focuses on non-elderly adults with co-morbidities. The manuscript was well written and comprehensive. Because of its length, it was however, challenging to read.
The introduction was quite lengthy with some text that was simply unnecessary for the "story" being told. Personally, this busy reviewer enjoys reading articles that get to the point much faster and stays on task to provide exactly what is needed to understand the study - no more or less.
This article uses a qualitative design which is not ever considered generalizable by those that use it regularly. Therefore, the sample size is not a limitation for generalizability, rather, the study's design is a limitation for generalizability. Given the approach for recruitment, only 14 were recruited with only 4 being labeled vaccine hesitant and only 2 of these not being vaccinated. Given the point of this work, one would think that the authors would oversample this group, but alas they did not.
Still, this reviewer found the labels (acceptor, cautious acceptor, etc.) very useful and the results overall useful though once again too lengthy. Keep in mind that I am not a social scientist, but a public health researcher so am unaccustomed to including so many details in a research article. This reviewer thinks the story can be told in a much more succinct way with less example quotes. Figure 2 seems unnecessary given Figure 3. They are overlapping. Figure 3 includes information on mandates which makes it better.
Section 3.5 Vaccine journeys was a very nice way to end the results section.
The discussion seemed to be a retelling of the results rather than a true synthesis of the results compared to the published literature. The limitations as stated seems inaccurate given the study design (see above). I agree with the conclusions overall.
I am sad that the authors didn't discuss socioeconomic status or other relevant demographic and educational levels of the people they interviewed. It's hard to know if their participants experienced financial hardship or not. Based on the quotes, I expect not. Could this be mentioned in the discussion?
Reviewer 3 Report
I would like to thank you for the opportunity to carry out this review.
[1] General. The study is well conducted with the right scientific rigor. The objectives are clearly stated, the methods are adequate, the presentation of the results is clear, and the discussion responds to the findings obtained. In addition, the conclusions respond to the research objectives.
[2] Introduction. The Introduction section is well written and gives all the information needed to understand the context of this study. The concept of vaccine hesitancy is well explained, and this is a strong point of the article's introduction.
Line 38. Could you please be more clear about what you mean with (1b)?
[3] Materials and Methods. The methods are described in sufficient detail to be fully understandable and to enable replication. The classification of the participants into five groups according to vaccine hesitancy is very interesting.
Line 211. What does group 1b mean?
[4] Results. The presentation of the results is clear.
[5] Discussion. The study findings are interpreted appropriately and within the context of existing evidence. Research limitations are well presented.
This paper is well written, and I enjoy reading through it.
